# Adipocytokines and Insulin Resistance: Their Role as Benign Breast Disease and Breast Cancer Risk Factors in a High-Prevalence Overweight-Obesity Group of Women over 40 Years Old

**DOI:** 10.3390/ijerph19106093

**Published:** 2022-05-17

**Authors:** Daniel Sat-Muñoz, Brenda-Eugenia Martínez-Herrera, Luis-Aarón Quiroga-Morales, Benjamín Trujillo-Hernández, Javier-Andrés González-Rodríguez, Leonardo-Xicotencatl Gutiérrez-Rodríguez, Caridad-Aurea Leal-Cortés, Eliseo Portilla-de-Buen, Benjamín Rubio-Jurado, Mario Salazar-Páramo, Eduardo Gómez-Sánchez, Raúl Delgadillo-Cristerna, Gabriela-Guadalupe Carrillo-Nuñez, Arnulfo-Hernán Nava-Zavala, Luz-Ma-Adriana Balderas-Peña

**Affiliations:** 1Departamento de Morfología, Centro Universitario de Ciencias de la Salud (CUCS), Universidad de Guadalajara (UdG), Cuerpo Académico UDG CA-874 “Ciencias Morfológicas en el Diagnóstico y Tratamiento de la Enfermedad”, 950 Sierra Mojada, Puerta 7, Edificio C, 1er Nivel, Guadalajara 44340, Mexico; 2Departamento Clínico de Oncología Quirúrgica, División de Oncología Hematología, UMAE, Hospital de Especialidades, Centro Médico Nacional de Occidente, Instituto Mexicano del Seguro Social, 1000 Belisario Domínguez, Guadalajara 44340, Mexico; 3Hospital General de Zona (HGZ), #02 c/MF “Dr. Francisco Padrón Puyou”, Órgano de Operación Administrativa Desconcentrada San Luis Potosi, IMSS, San Luis Potosi 78250, Mexico; bren.mtzh16@gmail.com; 4Unidad de Investigación Biomédica 02, UMAE Hospital de Especialidades (HE), Centro Médico Nacional de Occidente (CMNO), Instituto Mexicano del Seguro Social (IMSS), 1000 Belisario Domínguez, Guadalajara 44340, Mexico; luisquiroga@hotmail.com (L.-A.Q.-M.); javier.gonzalez5722@alumnos.udg.mx (J.-A.G.-R.); xicotencatl.gutierrez@alumno.udg.mx (L.-X.G.-R.); navazava@yahoo.com.mx (A.-H.N.-Z.); 5Facultad de Medicina, Universidad de Colima, Colima 28040, Mexico; trujillobenjamin@hotmail.com; 6Programa de Doctorado en Investigación Clínic, Coordinación de Posgrado, Centro Universitario de Ciencias de la Salud (CUCS), Universidad de Guadalajara (UdG), Guadalajara 44340, Mexico; 7Unidad Académica de Ciencias de la Salud, Clínica de Rehabilitación y Alto Rendimiento ESPORTIVA, Universidad Autónoma de Guadalajara, Zapopan 45129, Mexico; 8Carrera de Médico Cirujano y Partero, Coordinación de Servicio Social, Centro Universitario de Ciencias de la Salud (CUCS), Universidad de Guadalajara (UdG), Guadalajara 44340, Mexico; 9Carrera de Médico Cirujano y Partero, Coordinación de Servicio Social, Centro Universitario del Sur, Universidad de Guadalajara (UdG), Ciudad Guzmán 49000, Mexico; 10Comisión Interinstitucional de Formación de Recursos Humanos en Salud, Programa Nacional de Servicio Social en Investigación 2021, Demarcación Territorial Miguel Hidalgo, Ciudad de México 11410, Mexico; 11División de Investigación Quirúrgica, Centro de Investigación Biomédica de Occidente, Instituto Mexicano del Seguro Social, Órgano de Operación Administrativa Desconcentrada, Guadalajara 44340, Mexico; lealc36@yahoo.com.mx (C.-A.L.-C.); eportilla@mail.udg.mx (E.P.-d.-B.); 12Departamento Clínico de Hematología, División de Oncología Hematología, UMAE, Hospital de Especialidades, Centro Médico Nacional de Occidente, Instituto Mexicano del Seguro Social, 1000 Belisario Domínguez, Guadalajara 44340, Mexico; rubiojuradob@gmail.com; 13Academia de Inmunología, Departamento de Fisiología, Centro Universitario de Ciencias de la Salud (CUCS), Universidad de Guadalajara (UdG), 950 Sierra Mojada, Gate 7, Building O, 1st Level, Guadalajara 44340, Mexico; mario.sparamo@academicos.udg.mx; 14División de Disciplinas Básicas para Salud, Centro Universitario de Ciencias de la Salud (CUCS), Universidad de Guadalajara (UdG), 950 Sierra Mojada, Edificio N, Puerta 1, Planta Baja, Guadalajara 44340, Mexico; eduardo.gsanchez@academicos.udg.mx; 15Departamento Clínico de Radiologia e Imágen, Unidad Médica de Alta Especialidad (UMAE), Hospital de Especialidades (HE), Centro Médico Nacional de Occidente (CMNO,) Instituto Mexicano del Seguro Social (IMSS), 1000 Belisario Domínguez, Guadalajara 44340, Mexico; radelcri@outlook.com; 16Departamento de Morfología, Centro Universitario de Ciencias de la Salud (CUCS), Universidad de Guadalajara (UdG), Guadalajara 44340, Mexico; gabriela.cnunez@academicos.udg.mx; 17Unidad de Investigación Social Epidemiológica y en Servicios de Salud, Órgano de Operación Administrativa Desconcentrada, Guadalajara 44340, Mexico; 18Programa Internacional Facultad de Medicina, Universidad Autónoma de Guadalajara, Av. Patria 1201, Lomas del Valle, Zapopan 45129, Mexico; 19Servicio de Inmunología y Reumatología, División de Medicina Interna, Hospital General de Occidente, Secretaria de Salud Jalisco, Av. Zoquipan 1050, Zapopan 45170, Mexico

**Keywords:** adipocytokines, overweight-obesity, OR, insulin, HOMA IR, insulin resistance, benign breast disease (BBDS), breast cancer (BC), risk

## Abstract

Insulin levels, adipocytokines, and inflammatory mediators trigger benign breast disease (BBD) and breast cancer (BC). The relationship between serum adipocytokines levels, overweight-obesity, metabolic disturbs, and BC is unclear. Methods: To analyze the serum levels of the adipocytokines, insulin, and the HOMA IR in women without breast disease, with BBD or BC, and the role of these as risk factors for benign breast disease or breast cancer. Results: Adipsin values > 0.91 and visfatin levels > 1.18 ng/mL represent a risk factor to develop BBD in NBD lean women (OR = 18; and OR = 12). Data in overweight-obese women groups confirm the observation due to insulin levels > 2.6 mU/mL and HOMA IR > 0.78, with OR = 60.2 and 18, respectively; adipsin OR = 26.4, visfatin OR = 12. Breast cancer risk showed a similar behavior: Adipsin risk, adjusted by insulin and visfatin OR = 56 or HOMA IR and visfatin OR = 22.7. Conclusion: Adipose tissue is crucial for premalignant and malignant tissue transformation in women with overweight-obesity. The adipocyte–breast epithelium interaction could trigger a malignant transformation in a continuum, starting with BBD as premalignant disease, especially in overweight-obese women.

## 1. Introduction

Worldwide, breast cancer is a complex phenomenon characterized by coincident factors such as (1) a high prevalence, incidence, and mortality [1,2]; (2) the country-specific conditions, including deficiency in screening programs and limited access to treatment in some geographic areas [3]; and (3) the metabolic disturbances related to the high prevalence of overweight-obesity around the world, associated with hyperinsulinism, insulin resistance, metabolic syndrome [4], and adipocytokines altered levels [5]. Fortunately, those conditions are considered modifiable risk factors [6].

Adipocytokines leptin and adiponectin are related to BC progression. Simultaneously, the role of adipsin and visfatin is part of the altered metabolism into overweight-obesity, but actually, we have not fully understood the mechanisms associated with the development of benign breast disease (BBD) and breast cancer (BC).

Benign Breast Disease comprises various breast disorders identified through clinical data, mass in the breast, or imaging findings, without malignancy criteria. In the literature, premalignant diseases are considered risk markers by some authors [7]. The incidence peak of BBD occurs in child-bearing ages between 30 and 50 years. Breast pain (mastalgia) and fibrocystic changes represent approximately 50% of the BBD in clinical practice [8]. The risk of developing breast cancer is low, but some benign conditions represent the risk of breast cancer, such as complex cysts (23–32%), papillary lesions (16%), and radial scars (7%); in cases with doubt about the histological origin, confirmation should be obtained. So far, there have been few prospective analyses conducted to establish the actual risk of breast cancer in women with BBD [9].

Various reports describe that obese women have higher mortality rates associated with BC than lean women (1.8 to 2.2 times and the risk is 33% higher in overweight-obese women) [6,10]. The relationship between serum adipocytokine levels, overweight-obesity, metabolic disturbs, and BC [11,12] is one of the characteristics. The insulin levels and inflammatory mediators activated through adipocytokines trigger BBD development, considered a premalignant disease or dysplasia [13,14].

This paper analyzes the serum levels of the adipocytokines leptin, adiponectin, resistin, adipsin, and visfatin, as well as the insulin and HOMA IR in women without breast disease, with BBD or BC, and the role of those biochemical characteristics as risk factors of benign breast disease or breast cancer. This study’s base group constitutes naïve-treatment women identified through breast cancer screening programs.

## 2. Materials and Methods

This cross-sectional study was approved by the Research and Ethics Local Committee 1301. We included 246 women identified through mastographic or sonographic screening, confirming the pathologic diagnosis by biopsy in abnormal image results. According to the mastography and pathology results, we grouped the women as (1) no breast disease (NBD; n = 69, mammography or breast ultrasound BI-RADS I or II), (2) benign breast disease (BBD, n = 93), and (3) breast cancer (BC, n = 87). The three groups were subclassified by Body Mass Index (BMI) in lean (<25 kg/m^2^BS) and overweight-obesity (≥25 kg/m^2^BS) women.

We excluded women with previously known cancer, autoimmune diseases, chronic lung disease, cardiovascular diseases, renal failure, or contraindication to analyze body composition using electrical bio-impedance. All participants signed informed consent to participate.

We measured serum levels of adipocytokines, glucose, and insulin after calculating HOMA IR (HOMA index for insulin resistance).

Furthermore, we established the reference values based on the mean ±two standard deviations in women over 40 years old included in the NBD group, without chronic disease clinical data for the following values: HOMA IR, leptin, adiponectin, resistin, visfatin, and adipsin. For values with non-parametric behavior, we used median and percentile values of 2.5 and 97.5. The cut-off for normal values were HOMA IR = 0.78, insulin = 2.6 mU/mL, leptin = 27.5 ng/mL, adiponectin = 17.68 mg/mL, resistin = 0.59 ng/mL, visfatin = 1.18 ng/mL, and adipsin = 0.91 mg/mL (see Table 1).

### 2.1. Anthropometric and Body Composition Analysis

Anthropometric and body composition analyses were performed on all women with 8 h of fasting, with clean feet and no metal on the body. Height was measured with the SECA 213 stadiometer (SECA^®^, Hamburg-Wandsbek, Deutschland). Weight, body fat mass percentage, and skeletal muscle mass in kilograms (kg) were measured by bioelectrical impedance with a BF-601F device (TANITA^®^, Tokyo, Japan). To estimate the Body Mass Index (BMI), we divided the weight by the height squared.

### 2.2. Blood Sample

Glucose was determined in serum samples using the VITROS^®^ 350/5600 chemistry system from Ortho-Clinical Diagnostics (Johnson & Johnson, New Brunswick, NJ, USA). Five milliliters of venous blood were collected to obtain the serum. We processed the serum samples according to standard guidelines for insulin and adipocytokines (multi-test panel Bio-Plex Pro No. 171A7001M and Bio-Plex Pro 171A7003M, BIO-RAD^®^, Hercules, CA, USA). With the results of serum glucose and insulin concentrations, HOMA-IR was calculated, as described [15]. The diagnosis of insulin resistance applies to women with HOMA IR > 2.6 (values for Mexican women [16]).

### 2.3. Statistical Analysis

The statistical analysis comprises three phases. For descriptive analysis, the mean and standard deviation or median and interquartile range were calculated for biochemical values with non-parametric behavior.

Then, in proportions and percentages, we described conditions such as overweight-obesity, insulin resistance, dyslipidemia, and diagnosis groups. As part of the inferential analysis, we compared biochemical values by diagnostic groups and nutritional diagnosis (lean or overweight-obesity) with the ANOVA test and in each in-group with adjusted Bonferroni analysis to draw comparisons between groups. We performed the Kruskal–Wallis test with an adjusted Bonferroni analysis in the in-group analysis in variables with a non-parametric distribution. For differences between a specific percentage in one clinical condition, we calculated a chi-squared test, or Fisher’s exact test if the expected values were less than five.

Finally, the relationship between variables was identified using Pearson’s correlation (quantitative variables) or Spearman’s Rho (ordinal or relation between ordinal and quantitative variables); those with significance < 0.2 were selected and used in a logistic regression model to obtain the OR with 95% confidence intervals (CI95%). In the end, we constructed the model considering three disease conditions: (1) Benign breast disease risk in NBD women, (2) breast cancer risk in NBD women, and (3) breast cancer risk in BBD women; and two nutritional diagnoses: (A) the lean group and (B) the obese-overweight women group.

## 3. Results

According to BMI, 65 thin women (26.1%) and 184 overweight-obese women (73.9%) were identified, and each of them were subdivided according to their diagnosis in (a) NBD [n = 69; lean = 21 (31%) women, overweight-obesity = 48 (69%)]; (b) BBD [n = 93; lean = 18 (19.35%), overweight-obesity = 75 (80.25%)]; and BC [n = 87; lean = 26 (29.9%), overweight-obesity = 61 (70.1%)] (see Table 2).

### 3.1. Results by Disease and Nutritional Diagnosis

#### 3.1.1. Lean Women

We performed a sub-analysis comparing the NBD and BBD groups with each other and with each clinical stage. The BC group showed statistical differences in some biochemical markers (see Table 3 and Table 4): HDL cholesterol was higher in the BC CS II group (*p* = 0.010) compared to the BBD group. Insulin (*p* = 0.000) appeared with the lowest levels for the NDB group compared to BBD, BC CS-I, II, and III; the results were the same when comparing the BBD group with BC CS-III. HOMA IR (*p* = 0.000) showed biochemical behavior similar to serum insulin levels. Leptin (*p* = 0.007) and adiponectin (*p* = 0.028) reflected differences between the NBD and BBD and BC (CS I, II, and III) groups of women. There were no differences in resistin, visfatin, or adipsin values by CS in the BC group of women.

#### 3.1.2. Overweight-Obese Women

This sub-analysis aroused the following results (see Table 4): Age (*p* = 0.000) showed differences between NBD and BC CS-II. The BBD women group and women with BC in CS-I and II also reflected differences; BC CS III and BC CS I and II exhibited differences. The BMI (*p* = 0.034) described differences between NBD with BBD and BC (in each CS). Insulin (*p* = 0.000) reflected the lowest serum values for the NBD group, then BBD, BC CS I, II, and III, with no differences in BC CS IV.

HOMA IR (*p* = 0.000) showed biochemical behavior similar to serum insulin levels. Leptin (*p* = 0.000) had differences between the NBD, BBD, and BC CS-II group, and between BBD and BC CS-I and III, but no differences in BC CS IV. Adiponectin (*p* = 0.000) revealed significantly different serum levels in NBD to those in BBD, BC CS I, II, III, and IV, between BBD and BC CS IV, and between BC CS II and IV. Resistin (*p* = 0.000) showed statistical differences between the NBD group and BBD, BC CS I, and III. No distinctive pattern was observed in HDL-cholesterol, glucose, visfatin, or adipsin per CS in the BC group.

The analysis of anthropometric indicators revealed no significant differences between NBD, BBD, and BC in lean women. However, BMI was lower in the NBD group compared to the BBD and BC groups (see Table 4).

### 3.2. The Risk of BBD and BC Related to Metabolic and Biochemical Alterations Was Estimated with Logistic Regression and Odds Ratio (OR)

Metabolic syndrome was present in 81 women, 19 in the NBD group (with significant differences), 30 in the BBD group, and 32 in the BC group, with no significant differences (≥25 kg/m^2^). Therefore, the described condition did not represent a risk for the BBD or BC groups.

Adipsin values above 0.91 were directly related to the multivariate risk model, where we confirmed that visfatin levels above 1.18 ng/mL and adipsin levels above 0.91 mg/mL represent a risk factor in the development of BBD in lean women with BBD (OR 18, 95%CI: 1.9–163; and OR 12, 95%CI: 1.03–107; respectively), and when we adjust the risk of adipsin level above 0.91 mg/mL, for an elevated visfatin level, the risk increases (OR: 33 95%CI: 5.3–208).

The behavioral pattern is present in lean NBDs regarding the risk of developing BC: Elevated adipsin (OR: 18.5; 95%CI: 2–159.5), elevated visfatin (OR: 11.25; 95%CI: 1.3–98), and the adjusted model with elevated adipsin, adjusted for the visfatin level, represents an elevated risk of developing breast cancer, even in lean women (OR: 49.8; 95%CI: 8.2–304).

Data in the overweight-obese groups of women confirm the observation but emphasize the role of metabolic disturbance, which is indicated by insulin levels >2.6 mU/mL and HOMA IR > 0.78 (See Table 5). In this group of women with BMI > 25 kg/m^2^BS, the risks of developing BBD for the NBD group are insulin >2.6 mU/mL OR: 60.2 (CI95%: 8–460) and HOMA IR > 0.78 OR: 18 (CI95%: 4–79), adipsin > 0.91 OR: 26. 4 (CI95%: 6–117), visfatin OR 12 (CI95%: 2.7–53), adipsin risk-adjusted models for insulin and visfatin, and HOMA IR and visfatin showed higher risk values (OR: 75, CI95%: 18.8–301; and OR: 31.6, CI95%: 10–99).

## 4. Discussion

The frequency of overweight-obese and thin women found in our report is consistent with results reported by the National Health and Nutrition Survey in 2018 [17], with reported rates of overweight-obesity from 74.4% in women > 20 years to 88.1%, 84%, and 83.3% in women in their 40s, 50s, and 60s. These findings highlight two critical public health issues: The burden of overweight obesity [17] and the prevalence of breast cancer [1], the most common cancer in women in some geographic areas.

In recent years, adipose tissue has been considered an endocrine organ that synthesizes a variety of adipocytokines with pleiotropic effects on metabolism. These effects are related to carbohydrate and lipid metabolic pathways, inflammatory response, angiogenesis, and carcinogenesis [18].

In our current results, we can observe the absence of patients in clinical stage IV in the group of women with BMI < 25 kg/m^2^BS. Even though the etiology of BC is multifactorial, the inflammatory–anti-inflammatory factors balance, nutritional status, obesity, and genetic factors are involved, and their influence has been documented [19].

Advanced BC (III and IV) stages have enhanced the imbalance between inflammatory-anti-inflammatory molecules. The condition is perpetuated by obesity as a low-grade inflammatory state with metabolic dyshomeostasis, conditioning a higher probability of malignant transformation in the mammary epithelial tissue and disease progression. The above-described condition could be implied by the absence of clinical-stage IV cases in the lean women group and the observed differences in the studied metabolic markers [19].

The most-studied adipocytokines are leptin and adiponectin; however, molecules such as resistin, visfatin, and adipsin may influence the physiological and pathophysiological mechanisms described above, especially in overweight and obese subjects [16]; these, in turn, have an impact on the altered metabolic pathways seen in benign and malignant neoplastic breast disease [20,21,22]. Recently, Barone I et al. [23] described how the pandemic proportions of overweight and obesity represent a public health burden, with 1.9 billion overweight and 600 million obese adults worldwide suffering from one of these conditions.

In BBD and BC breast pathology, adipose tissue plays a central role coupled with adipocytokines such as leptin. Together, these perpetuate a sustained inflammatory and cell signaling environment, inflammation, angiogenesis, and metastasis, which could trigger the progression, incidence, prevalence, and mortality of BC [23], observations according to our results in this report with an elevated level of adipocytokines in women with BBD, with slightly decreased levels in women with BC, above the values of women without breast disease.

Authors such as Gallichio L. et al. identified the relationship of obesity with BBD and BC. In 2007, these researchers observed LEPR genes in Caucasian women with BBD, who developed BC in 6.1% (n = 61) associated with high BMI. The authors suggested that polymorphism in this gene may trigger the progression of BBD to BC when the gene is associated with excess body fat mass [14] (p. 2), recognizing the role of adipocytokines and their receptors on malignant transformation. Epidemiological studies reported the progression of BBD to BC is linked to atypical ductal and lobular hyperplasia and subsequent BC in a 4–5-fold and 3-fold ratio, respectively [24].

BBD is considered a risk factor for BC. If BBD is associated with elevated BMI and C-reactive protein levels, the risk increases through inflammatory pathways [25].

In the present study, we observed statistical differences between the NBD, BBD, and BC groups in the levels of adipocytokines, specifically leptin and adiponectin, and in metabolic indicators such as glucose, insulin, and HOMA IR for BBD in the lean and overweight-obese groups; and specifically, for the latter group, the significant differences were also for resistin and adipsin.

In this regard, we observed that women with BMI > 25 kg/m^2^ within the BC group showed differences in the biochemical behavior of leptin, adiponectin, resistin, and adipsin levels compared to women without breast disease. Differences in metabolic status were seen in insulin and HOMA IR levels.

BBD and progression to BC are related to overweight-obesity because it provides microenvironmental conditions associated with endocrine changes that promote tumor progression from benign to malignant features, loss of proliferative control, and invasion. Adipocytokines are essential molecular mediators in the obesity–BBD–BC pathophysiological axis [14] (p. 2), consistent with several research reports worldwide [26].

In the current results, we observed higher levels of leptin, adiponectin, and adipsin in the groups of BBD and BC compared to the NBD group.

Resistin levels were inversely associated with disease progression. Obese women without breast disease had the highest value (see Table 2 and Table 3), likely related to visceral fat production of resistin. Peripheral blood concentrations do not represent local adipocytokine concentrations in breast tissue [27]. The resistin production is elevated in obese experimental animal models. In their adipocytes, resistin decreases glucose transport in response to insulin, proposing a connection between obesity and insulin resistance [28]. In humans, macrophages produce resistin to proinflammatory mediators IL-1β, IL-6, and TNF-α. This metabolic feature implies the opposite effect of leptin and adiponectin [28].

We did not identify differences in resistin levels in the lean women group between NBD, BBD, and BC. However, in the group of overweight-obese women, resistin levels were elevated in the NBD group and decreased in the BBD and BC groups. In addition, these differences are present when analyzing the BC cases according to clinical stages (see Table 2 and Table 3).

Based on these results, we could hypothesize that the decreased resistin level in BBD may be a biochemical marker of metabolic alterations related to the development of BC and other neoplasms such as epithelial ovarian cancer, endometrial cancer, and esophageal squamous cell carcinoma [29,30,31].

Adipsin also showed significant differences. It is a molecule that limits the activation of the alternative complement pathway. In humans, macrophages and adipocytes express adipsin in obesity [32]. In BC, adipsin plays a crucial role in malignant transformation, invasion, and metastasis [27], but this role is not fully understood. We observed similar behavior in our patients, with significantly elevated adipsin levels in women with benign pathology and breast cancer, BMI > 25 kg/m^2^BS.

These findings and the results reported in vitro proliferation assays strongly suggest that adipsin produced in breast adipose tissue is part of the mechanisms involved in the inflammatory microenvironment, and the increased proliferation of cancer stem cells [33] in the crosstalk between tumor and breast stromal cells influences malignant transformation, disease progression, and patient prognosis [32,33] (p. 13).

The interaction of adipose cells with malignant transforming cells promotes the secretion of adipocytokines, and tumor growth increases their bioavailability as fuel for cancer cells. According to Nieman et al., molecules such as adipsin secreted by tumor-associated adipose cells may be driving the reduced efficacy of target therapy such as trastuzumab [34].

The adipsin/C3a pathway is essential for processes such as (1) immune regulation, (2) cell signaling, (3) cell migration, (4) insulin tolerance, (5) adipocyte differentiation, and (6) hematopoietic stem cell homing [35,36] (p.13). This pathway induces a chronic inflammatory process, an immunosuppressive tumor microenvironment, and angiogenesis. Goto et al. described complement activation and its association with tumor progression [32] (p. 13).

Molecular cascades that promote tumor progression through C3a include the activation of ERK 1/2 (extracellular signal-regulated kinase), the Akt pathway associated with IL-6 production, and VEGF extracellular matrix reorganization, promoting cellular invasion and migration [32] (p. 13).

Some studies suggest that adipsin/C3a could become a therapeutic target (lampalizumab-monoclonal antibody against adipsin), considering that adipsin is a mediator of the adipose–epithelial breast cell interaction that enhances proliferation and cancer stem cell properties of breast cancer cells [32] (p. 13).

Visfatin, in our results, showed a statistical correlation with the biochemical behavior of adipsin (data not shown). The origin of visfatin is visceral adipose tissue, and its expression has a solid correlation with visceral fat mass measured by tomography. Its metabolic function limits the synthesis of NAD involved in cellular and inflammatory pathways [18]. Its levels are elevated in obesity, diabetes mellitus 2, and metabolic syndrome, promoting atherosclerosis and endothelial dysfunction, and could be a biomarker in cancer [18] (p. 11).

Based on our previously published results and theoretical statements [37,38], we decided to develop several risk models based on the values we obtained in the lean women in the NBD group. We established mean values and upper cut-off points for insulin, HOMA IR, leptin, adiponectin, resistin, visfatin, and adipsin. We then dichotomized the variables as normal or abnormal. After calculating the estimated risk via the Odds Ratio (OR) and the 95% confidence interval (95% CI), we calculated the risk models for developing BBD and BC in the lean women in the NBD group and their overweight-obese counterparts.

The results are consistent with in vitro findings related to adipose-derived stem cells (ASCs) and their production of adipsin and leptin as part of fat-related abnormal proteins that could explain their different abilities to promote the growth of breast cancer tumor cells from surgical specimens. Moreover, ASC-mediated production of mesenchymal growth factors VEGF, TGF-β, and IL-6 is high.

Our current data show a paradoxical adiponectin response that could be explained by the recent reports published by Cerda-Flores RM et al. [39].

They describe the presence of ADIPOQ polymorphisms, specifically the SNPs rs2241766 (+45 T > G) and rs1501299 (+276 G > T). They observed that the rs1501299 (+276 G > T) polymorphism in a population from northeastern Mexico was associated with BC risk.

The ancestry of women from northern Mexico strongly suggests that elevated adiponectin levels are due to genetic risk markers linked to a non-functional molecule [39] (p. 15). Orozco-Argüelles L. et al. described the expression of adiponectin and its receptors ADIPOR 1 and 2 in postmenopausal breast cancer patients. After classifying women according to BMI, they described that overweight or obese women have lower adiponectin expression but higher ADIPOR1 expression in breast cancer tissue than in women with normal BMI, contrary to our findings [40]. These results could explain our findings: The elevated adiponectin levels probably induce receptor expression [41] (p. 15).

In overweight-obese women, the biological behavior of breast cancer is associated with high rates of invasion, metastasis, and recurrence. Adipocytes became the most abundant in breast tissue [41] (p. 15).

The CAAs’ (cancer-associated adipocytes) secretome profoundly affects the cellular and humoral response of the immune system to malignant cells [41,42] (p. 15). These effects act through feedback loops in adipocytokine regulation, metabolic reprogramming, extracellular matrix remodeling, and the presence of microRNAs.

Altered adipocytes overexpress inflammatory markers and proteases; their interaction with malignant breast cells may affect their form and function to promote proliferation, survival, migration, and invasion of breast cancer cells into neighboring tissues [41] (p. 15). All the situations described are present in our studied women with high rates of overweight-obesity.

## 5. Conclusions

Worldwide, breast cancer is the most common cancer among women. In Latin America, it is linked to high rates of overweight-obesity, one of the most important modifiable risk factors. Adipose tissue is crucial for premalignant and malignant tissues in women with high rates of overweight-obesity and a high prevalence of breast cancer.

The adipocyte–breast epithelium interaction could trigger malignant transformation on a continuum, starting with BBD as a premalignant disease, especially in overweight-obese women.

For the groups studied, adipsin values > 0.91 mg/mL, insulin levels above 2.6 mU/mL, HOMA IR > 0.78, and visfatin levels above 1.18 ng/mL posed a risk of developing BBD in women with NBD as high as OR:75 (*p* = 0.000) and for developing BC in women with NBD OR: 56 (*p* = 0.000). The only risk factor for developing cancer in the BBD group of women was postmenopausal status (OR: 2.12, CI95%: 1.07–4.4).

The main limitation of this study is temporality, as we cannot establish causality in cross-sectional studies. Therefore, it is necessary and desirable to realize a prospective follow-up of NBD, BBD, and BC populations to assess the behavioral pattern of adipocytokines as risk factors, calculating the relative risk or hazard ratio.

## Figures and Tables

**Table 1 ijerph-19-06093-t001:** Values for insulin, HOMA IR, and adipocytokines obtained in serum samples from 69 metabolic healthy Mexican women (reference values).

	Normal Values	Cutoff
Insulin (mU/mL)	1.15 (0.17–2.6)	>2.6
HOMA IR *	0.23 (0.03–0.78)	>0.78
Leptin (ng/mL)	1.99 (0.38–27.5)	>27.5
Adiponectin (μg/mL)	2.82 (0.64–17.68)	>17.68
Resistin (ng/mL)	3.9 (0.59–205.8)	<0.59
Visfatin (ng/mL)	0.84 (0.5–1.18)	>1.18
Adipsin (μg/mL)	0.47 (0.03–0.91)	>0.91

Homeostasis Model Assessment of Insulin Resistance *.

**Table 2 ijerph-19-06093-t002:** Anthropometric indicators in a group of over-40-year-old Mexican women undergoing mastographic screening.

	Women with BMI < 25				Women with BMI ≥ 25			
	No Breast Disease (n = 21)	Benign Breast Disease (n = 18)	Breast Cancer (n = 26)		No Breast Disease (n = 48)	Benign Breast Disease (n = 75)	Breast Cancer (n = 61)	
	Median Interquartile Interval (p25–p75)	Median Interquartile Interval (p25–p75)	Median Interquartile interval (p25–p75)	*p* *	Median Interquartile Interval (p25–p75)	Median Interquartile Interval (p25–p75)	Median Interquartile Interval (p25–p75)	*p* *
Age (years)	45 (41.5–63.5)	42.5 (33–51.7)	52 (44–58.3)	0.075	**52 (42.3–55)**	**49 (42–53) ***	**56 (47–61) ***	**0.003**
BMI (kg/m^2^)	23.6 (22–25)	23.2 (22–24.5)	22.8 (21.3–24)	0.089	**27.4 (26.6–31) ***	**29.2 (27.7–31)**	**30.3 (29–33.3) ***	**0.001**
Body Fat Percentage	32.4 (28.3–36)	33.8 (29.4–35)	31.6 (27.5–34)	0.363	38.5 (35.5–42)	39.3 (37.7–43)	41.4 (37.8–45)	0.061
Abdominal Fat Percentage	29.7 (23.7–33)	30.2 (24.3–30)	28.4 (24.4–31)	0.590	**34.5 (32–38) ***	**36.5 (34–42.5)**	**38 (33.8–42) ***	**0.026**
Total Muscle Mass	38.2 (35.2 40.8)	38.5 (36–40.6)	36.4 (34–38.6)	0.276	41 (39.5–43.5)	42 (38.7–44.7)	41.8 (39–46.6)	0.515
Skeletal Muscle Mass	16.2 (15–17.5)	16.2 (15–17)	15.4 (14–16.5)	0.310	17.3 (16.7–19)	17.9 (17–19.4)	18.2 (16.6–20)	0.357
Skeletal Muscle Mass Index (kg/m^2^)	6.3 (6.1–6.6)	6.2 (5.9–6.5)	6.2 (5.8–6.7)	0.270	7 (6.7–7.6)	7.3 (7–7.7)	7.4 (7.1–8.2)	0.098

Significant values in bold style. Body Mass Index *.

**Table 3 ijerph-19-06093-t003:** Biochemical and metabolic indicators in a group of over-40-year-old Mexican women undergoing mastographic screening.

	Women with BMI ** < 25				Women with BMI ** > 25			
	No Breast Disease (n = 21)	Benign Breast Disease (n = 18)	Breast Cancer (n = 26)		No Breast Disease (n = 48)	Benign Breast Disease (n = 75)	Breast Cancer (n = 61)	
	Median Interquartile Interval (p25–p75)	Median Interquartile Interval (p25–p75)	Median Interquartile Interval (p25–p75)	*p* *	Median Interquartile Interval (p25–p75)	Median Interquartile Interval (p25–p75)	Median Interquartile Interval (p25–p75)	*p* *
Leptin (ng/mL)	**1.5 (0.6–2.4) ***	**4.1 (2–8.6) ***	**3.03 (1.5–6.7) ***	**0.001**	**2.5 (1.4–4.1) ***	**6.5 (4–16.5) ***	**4.6 (2.4–11) ***	**0.000**
Adiponectin (μg/mL)	**4.4 (1.8–8.3) ***	**3.4 (1.2–14.4)**	**9.74 (7.3–18.2) ***	**0.007**	**2.5 (1.6–4.4) ***	**6 (2.5–10.2) ***	**7.7 (3–15.3) ***	**0.000**
Resistin (ng/mL)	1.9 (1–24.9)	1.7 (1.3–3.2)	1.7 (1.05–2.8)	0.594	**5.4 (1.3–28.2) ***	**2 (1.4–3) ***	**2.2 (1.5–3.5)**	**0.001**
Visfatin (ng/mL)	0.85 (0.77–0.9)	1.22 (0.57–2.1)	0.85 (0.65–1.99)	0.689	0.87 (0.78–0.98)	0.91 (0.7–1.6)	0.84 (0.65–2)	0.714
Adipsin (μg/mL)	0.35 (0.25–0.48)	0.24 (0.02–1.3)	0.9 (0.00–1.3	0.254	**0.47 (0.4–0.56) ***	**1.04 (0.09–1.5) ***	**0.9 (0.06–1.3)**	**0.009**
Glucose (mg/dL)	**80 (76–88) ***	83.5 (76.3–89)	**88 (81–105.3) ***	**0.050**	89 (81–96.7)	88 (80–99)	87 (82–101)	0.784
Insulin (μU/mL)	**1.02 (0.63–1.2) ***	**2.1 (1.3–3.6) ***	**3 (1.8–5.6) ***	**0.000**	**2 (0.77–1.6) ***	**3.2 (1.8–5.2) ***	**2.8 (1.6–4.6) ***	**0.000**
HOMA-IR ***	**0.22 (0.13–0.3) ***	**0.45 (0.3–0.8) ***	**0.61 (0.4–2.2) ***	**0.000**	**0.3 (0.16–0.36) ***	**0.67 (0.41–1.2) ***	**0.66 (0.37–1) ***	**0.000**
	**n (%)**	**n (%)**	**n (%)**	** *p* **	**n (%)**	**n (%)**	**n (%)**	** *p* **
Insulin Resistance (HOMA IR *** > 2.6)	0 (0%)	1 (5.6%)	5 (9.2%)	0.063	0 (0%)	11 (14.7%)	6 (9.8%)	0.023

Significant values are in bold *. Body Mass Index **, Homeostasis Model Assessment of Insulin Resistance ***.

**Table 4 ijerph-19-06093-t004:** Biochemical and metabolic indicators in a group of over-40-year-old Mexican women undergoing mastographic screening.

**Women with BMI ^ƒ^ < 25**							
	**NBD (n = 21)**	**BBD (n = 18)**	**BC (n = 26)** **Median Interquartile Interval (p25–p75)**				
	**Median Interquartile Interval (p25–p75)**	**Median Interquartile Interval (p25–p75)**	**CS I**	**CS II**	**CS III**	**CS IV**	** *p* ** *****
Age (years)	45 (41–63)	42.5 (32.75–51.7)	52 (45.7–55.5)	49.5 (39.3–58.7)	47 (41.25–62)	----	0.139
BMI (kg/m^2^SC)	23.7 (22.4–24.8)	23.2 (22.2–24.5)	22.4 (21.4–23.5)	22.3 (20.1–23.8)	23.2 (21.1–24)	----	0.139
Leptin (ng/mL)	**1.5** **(0.6–2.9)**	**4.1** **(1.9–8.6)**	**3.8** **(1.8–6.1)**	**3** **(1.7–8.1)**	**2.3** **(1–6.6)**	**-----**	**0.007**
Adiponectin (μg/mL)	**4.4** **(1.8–7.9)**	**3.4** **(1.2–14.4)**	**9.7** **(5.5–12.8)**	**10.1** **(8.6–28.7)**	**9.5** **(5.8–29)**	**-----**	**0.028**
Resistin (ng/mL)	1.9 (1–25)	1.7 (1.4–3.1)	1.5 (0.9–3.1)	1.5 (1.2–2.3)	2.3 (1.1–4.2)	-----	0.721
Visfatin (ng/mL)	0.9 (0.8–0.9)	1.2 (0.6–2.1)	0.6 (0.5–2.4)	1.0 (0.8–1.3)	0.7 (0.6–2.5)	-----	0.521
Adipsin (μg/mL)	0.35 (0.3–0.5)	0.24 (0.02–1.3)	0.7 (0.2–1.1)	0.9 (0.2–1.3)	1.1 (0.0–1.9)	-----	0.544
HDL-cholesterol (mg/dL)	**54** **(38–63)**	**47** **(27.8–47.25)**	**49.5** **(47–54.7)**	**51** **(37–65.5)**	**47** **(31.2–47.7)**	**----**	**0.010**
Glucose (mg/dL)	**82** **(76–89)**	83.5 (76.3–89)	**85.5** **(81–104.3)**	**86** **(78–7-98)**	**95.5** **(82.5–118)**	**-----**	0.132
Insulin (μU/mL)	**1.03** **(0.7–1.2)**	**2.1** **(1.3–3.6)**	**2.5** **(1.7–19.6)**	**2.5** **(1.1–4.6)**	**4.1** **(2.6–7.4)**	**-----**	**0.000**
HOMA-IR ^∞^	**0.22** **(0.1–0.3)**	**0.45** **(0.3–0.8)**	**0.45** **(0.4–4.1)**	**0.5** **(0.3–0.9)**	**1** **(0.6–2.7)**	**------**	**0.000**
Independent-samples Kruskal-Wallis Test	* Observed differences: (a) HDL-cholesterol: NBD-BBD; BBD-CSII (b) Insulin: NBD-BBD; NBD-CS I, CS II, CS III (c) I HOMA IR: NBD-BBD; NBD-CS I, CS II, CS III and BBD-CS III (d) Leptin: NBD-BBD; NBD-CS I, CS II and CS II (e) Adiponectin: NBD-CS I, CS II and CS III						
**Women with BMI ^ƒ^ ≥ 25**							
	**NBD (n = 48)**	**BBD (n = 75)**	**BC (n = 61)** **Median Interquartile Interval (p25–p75)**				
	**Median Interquartile Interval (p25–p75)**	**Median Interquartile Interval (p25–p75)**	**CS I**	**CS II**	**CS III**	**CS IV**	** *p* ** *****
Age (years)	**52** **(45.7–55)**	**49** **(42–53)**	**58** **(45–63)**	**58** **(48–66)**	**48** **(38.7–52.7)**	**57** **(52–57)**	**0.000**
BMI (kg/m^2^SC)	**27.4** **(26.6–30.8)**	**29.2** **(27.7–31.2)**	**33** **(28.6–36.9)**	**30.2** **(28.6–32.8)**	**30.3** **(28.6–32.6)**	**29** **(28.9–29)**	**0.034**
Leptin (ng/mL)	**2.5** **(1.4–4.1)**	**6.5** **(4.2–16.5)**	**4.5** **(1.2–10.7)**	**6.1** **(2.9–12.9)**	**2.9** **(2.2–5.9)**	**3.1** **(3–3)**	**0.000**
Adiponectin (μg/mL)	**2.5** **(1.6–4.4)**	**5.9** **(2.5–10.2)**	**7.7** **(2.1–16.7)**	**8.6** **(2.7–11.7)**	**4.7** **(3.1–17.5)**	**22.9** **(11.7–23)**	**0.000**
Resistin (ng/mL)	**5.4** **(1.4–28.2)**	**2** **(1.4–2.3)**	**1.8** **(0.7–2.4)**	**2.6** **(1.6–3.8)**	**1.7** **(1.5–2.3)**	**2.4** **(2–3.2)**	**0.000**
Visfatin (ng/mL)	0.9 (0.8–1)	0.9 (0.7–1.6)	0.84 (0.5–2.2)	1.11 (0.7–2.2)	0.7 (0.6–0.9)	0.9 (0.6–0.9)	0.148
Adipsin (μg/mL)	0.5 (0.4–0.6)	1 (0.1–1.5)	0.9 (0.1–1.5)	0.7 (0.02–1.3)	0.83 (0.15–1.5)	1.3 (0.2–1.3)	0.082
HDL-cholesterol (mg/dL)	48.5 41–59.3	47 41–47	47 46–47	47 40–47	47.5 42.5–57.5	46 42–46	0.349
Glucose (mg/dL)	89 (81–97.3)	88 (80–99)	98 (84–112)	87 (82–97)	83.5 (81–94)	100 (86–100)	0.497
Insulin (μU/mL)	**1.2** **(0.8–1.5)**	**3.2** **(1.8–5.2)**	**2.8** **(1.5–14)**	**3.4** **(1.8–5)**	**2.7** **(1.3–3.4)**	**1.6** **(1.5–1.6)**	**0.000**
HOMA-IR	**0.3** **(0.1–0.4)**	**0.7** **(0.4–1.2)**	**0.61** **(0.8–1.5)**	**0.7** **(0.4–4.1)**	**0.5** **(0.4–1.1)**	**0.4** **(0.3–0.8)**	**0.000**
Independent-samples Kruskal-Wallis Test	* Observed differences: (a) Age: NBD-CS II; BBD-CS I, and II; CS III-CS I, and CS II (b) BMI: NBD-BBD; NBD-CS I, CS II, and CIII (c) Insulin: NBD-BBD; NBD-CS I, CS II, CS III (d) HOMA IR: NBD-BBD; NBD-CS I, CS II, anIS III (e) Leptin: NBD-BBD; NBD-CS II; BBD-CS I and CS III (f) Adiponectin: NBD-BBD; NBD-CS I, CS II, CS III and CS IV; BBD-CS IV; CS III-CS IV (g) Resistin: NBD-BBD; NBD-CS I, and CS III						

* Significant values are in bold. Body Mass Index ^ƒ^. Homeostasis Model Assessment of Insulin Resistance ^∞^.

**Table 5 ijerph-19-06093-t005:** Estimated risk to develop benign breast disease and breast cancer in Mexican women (OR calculated through logistic regression).

**(A) Estimated Risk to Develop Benign Breast Disease in Women without Breast Disease**					
**Women with BMI** **^ƒ^ < 25**			**Women with BMI** **^ƒ^ ≥ 25**		
**Raw risk models**	**OR (IC95%)**	***p* ***	**Raw risk models**	**OR (IC95%)**	***p* ***
Visfatin > 1.18	18 (1.9–163)	0.010	Insulin > 2.6	60.2 (8–460)	0.000
Adipsin > 0.91	12 (1.3–107)	0.030	Adipsin > 0.91	26.4 (6–117)	0.000
			HOMA IR ^∞^ > 0.78	18 (4–79)	0.000
			Visfatin	12 (2.7–53)	0.001
			**Adjusted risk model 1**	**OR (IC95%)**	***p* ***
			Adipsin > 0.91 adjusted by Insulin > 2.6 and Visfatin > 1.18	75 (18.8–301)	0.000
**Adjusted risk model 1**	**OR (IC95%)**	***p* ***	**Adjusted risk model 2**	**OR (IC95%)**	***p* ***
Adipsina > 0.91 adjusted by Visfatin > 1.18	33.3 (5.3–208)	0.000	Adipsin > 0.91, adjusted by HOMA IR ^∞^ > 0.78, and visfatin > 1.18	31.6 (10–99)	0.000
**(B) Estimated risk to develop breast cancer in women without breast disease**					
**Raw risk models**	**OR (IC95%)**	***p* ***	**Raw risk models**	**OR (IC95%)**	***p* ***
Adipsin > 0.91	18.5 (2–159.5)	0.008	Insulin > 2.6	71.5 (9.2–555)	0.000
Visfati > 1.18	11.25 (1.3–98)	0.029	Adipsin > 0.91 8	21.5 (4.7–97)	0.000
			Visfatin > 1.18	13 (2.9–59)	0.001
			HOMA IR ^∞^ > 0.78	12 (2.6–55)	0.001
**Adjusted risk model 1**	**OR (IC95%)**	***p* ***	**Adjusted risk model 1**	**OR (IC95%)**	***p* ***
Adipsin > 0.91 ajusted by Visfatin > 1.18	49.8 (8.2–304)	0.000	Adipsin > 0.91 adjusted by Insulin > 2.6 and by Visfatin > 1.18	56 (14–227)	0.000
			**Adjusted risk model 2**	**OR (IC95%)**	***p* ***
			Adipsin > 0.91, adjusted by HOMA IR ^∞^ > 0.78, and visfatin > 1.18	22.7 (7.3–70)	0.000
**(C) Estimated risk to develop breast cancer in women with benign breast disease**					
**Raw risk models**	**OR (IC95%)**	***p* ***	**Raw risk models**	**OR (IC95%)**	***p* ***
Insulin > 2.6	2.4 (0.7–8.3)	0.152	Hormonal status (menopausal)	2.2 (1.07–4.4)	0.032

* Significant values are in bold. Body Mass Index ^ƒ^. Homeostasis Model Assessment of Insulin Resistance ^∞^.

## Data Availability

The datasets generated and/or analyzed during the current study are not publicly available because they are the property of the Instituto Mexicano del Seguro Social. Institutional and federal dispositions restrict unlimited access to personal data, but they are available from the corresponding author on reasonable request with prior authorization from the institution.

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
