# Peer review of "Adipocytokines and Insulin Resistance: Their Role as Benign Breast Disease and Breast Cancer Risk Factors in a High-Prevalence Overweight-Obesity Group of Women over 40 Years Old"

_ijerph, 2022, doi:10.3390/ijerph19106093_

Round 1
Reviewer 1 Report
This article entitled “Adipocytokines and Insulin Resistance; Their Role as Benign Breast Disease and Breast Cancer Risk Factors in a High-Prevalence Overweight-Obesity Group of Women over 40 Years Old” provides evidences of serum analysis to prove that the continuous interactions between adipocyte and breast epithelium could trigger malignant transformation of breast tissue in obese women. It is an interesting study but remaining some criticisms.
- What is the definition of benign breast disease (BBD)? And, how/what rate does BBD develop to breast? The authors should give details in Introduction.
- In the enrolled individuals, they were separated into three independent groups, normal, BBD, BC. It can not be linked the BBD developing to BC.
- In Table 4, why the level of the parameters, such as BMI, leptin, insulin, and HOMA-IR, in CS IV of BC showed lower than the other groups? It should be discussed.
- In Table 4, there is no CS IV appeared in the subgroup of BMI<25. Does obesity influence the severity of BC? It should be discussed.
- In Discussion, some paragraphs should be reorganized. e.g. line 309-326, all the sections mentioned resistin should be put together.
- Some typos should be proofread.
Author Response
Dear reviewer we add in the attached file our response to your comment and the way that we answer to your requirements
Thanks in advance for your useful comments

Reviewer 2 Report
The authors state the existence of statistical differences between the NBD, BBD and BC groups for the levels of various adipocytokines and metabolic indicators in lean and overweight/obese women.
- Line 240: BPD. BPD or BBD? It looks like a mistake.
- Line394: CAA. Please describe this abbreviation for the first time. I assume you mean "Adipocyte-Associated Cancer".
- There is literature that does not appear in the text. References: 21, 22, 35, 36.
- Table 4. Why is there no data in the CS IV column?
- Abstract (lines 87 – 88): Do you repeat the adipsin values twice? The authors should also indicate that NBD are your controls without cancer.
- Lines 137 – 139: “After calculating HOMA IR (HOMA index for insulin resistance), we measured serum levels of adipocytokines, glucose, and insulin after calculating HOMA IR (HOMA index for insulin resistance)”. This paragraph is very redundant.
- Line 212: Please, I do not see figure 1.
- Lines 325 – 326: “Based on these results, we could hypothesize that the decreased resistin level in BBD may be a biochemical marker of metabolic alterations related to the development of BC”.
It seems excellent to me, are there indications with other types of cancer that can confirm it?
The authors should conclude their discussion with limitations and future prospects of this study.
Author Response
In the attached file we answer to your comments. We highlighted the changes in the letter and in the article file.
Thanks in advance for your valuable comments

Round 2
Reviewer 2 Report
I appreciate the improvements to the manuscript and the clarifications provided.